# Unaltered Liver Regeneration in Post-Cholestatic Rats Treated with the FXR Agonist Obeticholic Acid

**DOI:** 10.3390/biom11020260

**Published:** 2021-02-10

**Authors:** Lianne R. de Haan, Joanne Verheij, Rowan F. van Golen, Verena Horneffer-van der Sluis, Matthew R. Lewis, Ulrich H. W. Beuers, Thomas M. van Gulik, Steven W. M. Olde Damink, Frank G. Schaap, Michal Heger, Pim B. Olthof

**Affiliations:** 1Jiaxing Key Laboratory for Photonanomedicine and Experimental Therapeutics, Department of Pharmaceutics, College of Medicine, Jiaxing University, Jiaxing 314001, Zhejiang, China; dehaan.lianne@gmail.com; 2Department of Surgery, Amsterdam UMC, Location AMC, University of Amsterdam, 1105 AZ Amsterdam, The Netherlands; t.m.vangulik@amsterdamumc.nl (T.M.v.G.); p.olthof@erasmusmc.nl (P.B.O.); 3Department of Pathology, Amsterdam UMC, Location AMC, University of Amsterdam, 1105 AZ Amsterdam, The Netherlands; j.verheij@amsterdamumc.nl; 4Department of Gastroenterology and Hepatology, Leiden University Medical Center, 2333 ZA Leiden, The Netherlands; rowan.vg@gmail.com; 5National Phenome Centre, Department of Metabolism, Digestion and Reproduction, Imperial College London, London W12 0NN, UK; v.horneffer@imperial.ac.uk (V.H.-v.d.S.); matthew.lewis@imperial.ac.uk (M.R.L.); 6Department of Gastroenterology & Hepatology and Tytgat Institute for Liver and Intestinal Research, Amsterdam Gastroenterology & Metabolism, Amsterdam UMC, Location AMC, 1105 AZ Amsterdam, The Netherlands; u.h.beuers@amsterdamumc.nl; 7Department of Surgery & NUTRIM School of Nutrition and Translational Research in Metabolism, Maastricht University Medical Center, 6200 MD Maastricht, The Netherlands; steven.oldedamink@maastrichtuniversity.nl (S.W.M.O.D.); frank.schaap@maastrichtuniversity.nl (F.G.S.); 8Department of General, Visceral and Transplantation Surgery, RWTH University Hospital Aachen, 52074 Aachen, Germany; 9Department of Pharmaceutics, Utrecht Institute for Pharmaceutical Sciences, Utrecht University, 3584 CG Utrecht, The Netherlands; 10Department of Surgery, Erasmus Medical Center, 3015 GD Rotterdam, The Netherlands

**Keywords:** bile duct obstruction, pharmacological intervention, bile salts, liver regeneration, partial hepatectomy, biliary obstruction, basolateral and canalicular transporters, bile acid metabolism

## Abstract

In a previous study, obeticholic acid (OCA) increased liver growth before partial hepatectomy (PHx) in rats through the bile acid receptor farnesoid X-receptor (FXR). In that model, OCA was administered during obstructive cholestasis. However, patients normally undergo PHx several days after biliary drainage. The effects of OCA on liver regeneration were therefore studied in post-cholestatic Wistar rats. Rats underwent sham surgery or reversible bile duct ligation (rBDL), which was relieved after 7 days. PHx was performed one day after restoration of bile flow. Rats received 10 mg/kg OCA per day or were fed vehicle from restoration of bile flow until sacrifice 5 days after PHx. Liver regeneration was comparable between cholestatic and non-cholestatic livers in PHx-subjected rats, which paralleled liver regeneration a human validation cohort. OCA treatment induced ileal *Fgf15* mRNA expression but did not enhance post-PHx hepatocyte proliferation through FXR/SHP signaling. OCA treatment neither increased mitosis rates nor recovery of liver weight after PHx but accelerated liver regrowth in rats that had not been subjected to rBDL. OCA did not increase biliary injury. Conclusively, OCA does not induce liver regeneration in post-cholestatic rats and does not exacerbate biliary damage that results from cholestasis. This study challenges the previously reported beneficial effects of OCA in liver regeneration in cholestatic rats.

## 1. Introduction

The liver is key to homeostasis and metabolism. As the gatekeeper of circulation, the liver filters portal venous blood and prevents toxic substances derived from the intestines from entering systemic circulation. Moreover, the liver controls bile acid (BA) homeostasis through production and biliary secretion as well as reabsorption and secondary metabolism. In addition, the liver has the capacity to regenerate. After hepatic injury or surgical resection, the liver rapidly regenerates to a predefined liver:body weight ratio to balance its metabolic function [1]. This equilibration to a preset ideal liver size is often referred to as the “hepatostat” [2]. Although liver function is optimal at this predefined liver:body weight ratio, up to 70% of the liver can be removed without noteworthy disturbance of homeostatic, synthetic, and detoxifying function [3,4,5].

Partial removal of the liver, or partial hepatectomy (PHx), is often the only curative treatment option for patients with hepatobiliary malignancies such as perihilar cholangiocarcinoma. This tumor arises from the biliary tract between the segmental bile ducts and cystic duct and often causes obstructive cholestasis and jaundice [6,7,8,9]. Cholestasis induces liver damage with systemic consequences [10,11] and hampers the regenerative capacity of the liver [12]. Consequently, extended resections in cholestatic livers are associated with high morbidity and mortality [13]. To minimize post-PHx mortality, a variety of invasive procedures has been devised to relieve cholestasis and to increase future remnant liver size. Nevertheless, PHx in patients with perihilar cholangiocarcinoma is still associated with a 90-day mortality rate of up to 18% [14].

Considering the high mortality rate, recent studies have investigated pharmacological modulation of liver regeneration to augment future remnant liver size [15,16,17,18,19,20,21]. In this context, an important role has been ascribed to BAs and drugs that activate BA receptors, most notably the farnesoid X receptor (FXR) [1,22,23,24]. BAs can directly (through hepatic FXR) and indirectly (through intestinal FXR and the subsequent induction and portal release of fibroblast growth factor (FGF)15/19 (rat/human orthologue)) trigger hepatocyte proliferation via the portal circulation [22]. A detailed summary of the molecular pathways controlling BA signaling through FXR has been recently published [1]. Obeticholic acid (OCA) is a synthetic, potent, and selective FXR agonist that is already approved for reducing cholestasis in patients with primary biliary cholangitis (PBC) [25,26]. Moreover, OCA can be used to study the effects of FXR on liver regeneration [24,27].

Oral OCA for 7 days induced liver growth in cholestatic rats [27]. These results suggest that OCA has the ability to stimulate liver growth under conditions of cholestasis and liver inflammation [11], which might improve the surgical starting point for major liver resection and thereby reduce mortality. Although OCA induced liver growth, it also exacerbated biliary injury, which was ascribed to the induction of the canalicular bile salt export pump (BSEP) and subsequent forced counter-gradient pumping of BAs into an obstructed biliary tree. Hepatobiliary injury markers normalized after restoration of hepatic bile flow under continued OCA treatment, suggesting that OCA mainly aggravates biliary injury during obstructive cholestasis. Compensatory, cytoprotective mechanisms that had been observed in vitro [28] were not engaged in vivo under these conditions. Comparable adverse events––including increased alanine transaminase (ALT), hyperbilirubinemia, and abnormal liver function tests––were also reported in patients with PBC, albeit with low incidence rates [25,26].

Although this study yielded promising results for the use of OCA in stimulating liver growth, the study setting was not fully representative of the clinical situation. Whereas the cholestatic animals in the initial study underwent PHx and biliary drainage simultaneously, biliary drainage in patients with perihilar cholangiocarcinoma is normally performed several weeks before surgery [29,30]. Therefore, the results from this study could provide an inaccurate representation of the real pro-regenerative effects of OCA in the clinical setting.

In the present study, an animal model that better reflects the clinical situation was used. Based on the pharmacological properties of OCA, it was hypothesized that OCA would induce hepatocyte proliferation and liver (re)growth in both post-cholestatic and non-cholestatic rats through the well-established FXR-mediated molecular signaling cascades. In the modified animal model, bile flow was therefore restored one day prior to PHx and OCA was administered 1 day after BDL reversal. The aim was to confirm FXR signaling functionality in an altered hepatopathological landscape while ruling out the influence of biliary injury on these processes caused by the combination of OCA treatment and obstructive cholestasis. It was expected that liver (re)growth would be unimpaired as in our previous study and that OCA-mediated biliary damage would be alleviated.

## 2. Materials and Methods

A list of abbreviations is provided in the Appendix A. Appendix A are referenced with the prefix ‘S.’

### 2.1. Animals

Male Wistar rats (*n* = 108, mean weight 278 g, range 231–352 g) were purchased from Harlan (Horst, The Netherlands) and acclimated for at least 1 week. The animals were housed in groups under standardized laboratory conditions and received a regular chow diet (Harlan Teklad, Madison, WI, USA) and water *ad libitum*.

All experimental protocols were approved by the animal institutional review board (Amsterdam UMC, location AMC, protocol #BEX183AB, approved on 23 March 2016) and were performed in compliance with institutional guidelines and the *National Institute of Health Guidelines for the Care and Use of Laboratory Animals* (8th edition). The animal experiments are reported according to the ARRIVE guidelines.

### 2.2. Experimental Design

Three hours prior to surgery, rats were administered 0.06 mg/kg of buprenorphine subcutaneously for analgesic care. Anesthesia was induced and maintained using a mixture of air:O_2_ (1:1) and 2–3% isoflurane (Forene, Abbott Laboratories, Chicago, IL, USA) [28]. A heating pad and infrared lamp were used to maintain body temperature at 37.0 ± 0.2 °C.

Rats were randomly assigned to one of 6 groups (Figure 1). Two cholestasis groups, a group that underwent reversible bile duct ligation only (BDL, *n* = 6), and a baseline group that received sham surgery (Sham, *n* = 6) were sacrificed 1 day before PHx (*t* = −1 day). The PHx groups comprised a control group (Sham-Veh, *n* = 24), a post-obstructive cholestasis group not given OCA (BDL-Veh, *n* = 24), a group that received OCA and sham surgery (Sham-OCA, *n* = 24), and a post-obstructive cholestasis group that was given OCA (BDL-OCA, *n* = 24). In total, 10 rats died due to complications of cholestasis or surgery.

Reversible BDL (rBDL) was performed with a polyethylene catheter connected to silastic tubing with a closed distal tip, which was inserted into the extrahepatic part of the common bile duct [31]. Control animals underwent abdominal surgery and bile duct mobilization but without the bile duct ligation and cannulation. The duration of obstructive cholestasis was 7 days.

The bile flow into the intestines was restored by removing the closed cannula tip and inserting the cannula into the duodenum [31]. rBDL was reliability accounted for by measuring serum BA and bilirubin levels on *t* = −1 day and on day 0. Control rats were subjected to a second sham procedure. After the second surgery, rats received daily oral gavage of either 10 mg/kg (based on earlier studies [24,28,32,33]) OCA (Intercept Pharmaceuticals, New York, NY, USA) in MilliQ water containing 1% methylcellulose (1 mL per 300 g body weight) or an equivalent dose of 1% methylcellulose in MilliQ as a vehicle control administered at a set time point each morning.

One day after alleviation of the rBDL or the second sham surgery, animals underwent 70% PHx. In brief, rats were anesthetized as described above and a midline laparotomy was performed. The liver was mobilized and the median and lateral lobes were resected [34]. Rats continued to receive daily oral gavage of OCA or vehicle control from 1 day before PHx until sacrifice. All rats were weighted daily from the start of the experiment (*t* = −8 days) until the moment of sacrifice (*t* = 5 days) as part of standard animal health monitoring and the risk of excessive weight loss [28].

The rats were sacrificed in the morning of day 5 by exsanguination via cardiac puncture under 2–3% isoflurane anesthesia directly before PHx (*t* = 0 d, *n* = 6/group) and on *t* = 1 day (*n* = 6/group), 3 days (*n* = 6/group), and 5 days (*n* = 6/group) after PHx. In total, 10 rats (*n* = 6 (BDL-Veh), *n* = 1 (Sham-OCA), and *n* = 3 (BDL-OCA)) were sacrificed prematurely due to intra-operative bile duct destruction or post-operative bile leakage. Blood samples were collected in EDTA- or heparin-anticoagulated Vacutainers (BD Biosciences, Franklin Lakes, NJ, USA). The ileum and liver were excised following a midline laparotomy and weighed. Left lateral liver sections and the last 2 cm of the terminal ileum were stored in RNAlater (Qiagen, Venlo, The Netherlands) for RNA isolation, and median liver lobe sections were fixed in 10% formalin for histology. The remaining liver sections were snap-frozen in liquid nitrogen for the determination of the intrahepatic BA composition and stored at −80 °C until further analysis.

The regenerated liver mass was used to calculate the percentage of liver regrowth compared to the total liver mass before PHx (*t* = −1 day). The resected liver segments were used to estimate the remnant liver mass, where the resected segments represent 70% of the total liver mass [32]. Liver weight was expressed as a percentage of total body weight. To correct for hepatic water content, dry liver weight was determined. The dry liver weight:body weight ratio was used as an outcome measure for liver regrowth.

### 2.3. Indocyanine Green Liver Function Test

Analysis of liver function was performed with indocyanine green (ICG) as described previously [33,35]. ICG is taken up by liver-specific transporters [36] and serves as a comprehensive liver function test [37,38]. One day after PHx (*t* = 1 day), 2.5 mg/kg of ICG (Pulsion Medical Systems, München, Germany) in sterile water (2.5 mg/mL) was injected into the penile vein. Blood (1 mL) was collected from the tail vein at 1, 2, 3, 4, 6, 8, and 10 min after ICG injection. Blood samples were centrifuged at 10,000× *g* for 10 min at room temperature (RT) [39], and 150 µL of platelet-poor plasma was diluted with 750 µL of 0.9% NaCl containing 1% (*w*/*v*) fetal bovine serum and 0.9% NaCl in water. ICG absorption was measured at 805 nm in a plate reader (Synergy HT, BioTek Instruments, Winooski, VT, USA). Elimination half-life (t_½_) was calculated from the slope of the semi-logarithmic decay curve. The plasma disappearance rate (ICG-PDR) was calculated using the formula $ICG-PDR=100\times \frac{ln2}{t_{1/2}}$ and expressed as %/min [40]. Liver function was expressed as mean ± SEM PDR.

### 2.4. Histology

Formalin-fixed liver sections were immersed in graded steps (70%, 80%, 90%, and 100%) of ethanol and xylene for dehydration and were embedded in paraffin. Five-µm thick sections were stained with hematoxylin and eosin (H & E) to assess inflammation, fibrosis, and necrosis [11]. Semi-quantitative analysis using a scoring system (Appendix A) [28] was performed by an expert hepatopathologist (JV) blinded to the group allocation and study design. The number of mitotic hepatocytes observed in a high-power field (HPF; 400× magnification) with the highest degree of mitotic activity was used as outcome measure for hepatocyte proliferation.

### 2.5. Clinical Chemistry

Heparin-anticoagulated whole blood was centrifuged at 3,000× *g* for 10 min at RT. Platelet-poor plasma samples were analyzed for bilirubin, gamma-glutamyltransferase (gGT), alkaline phosphatase (ALP), alanine transaminase (ALT), and aspartate transaminase (AST) levels using a Cobas 8000 modular analyzer (Roche, Basel, Switzerland) at the Department of Clinical Chemistry (Amsterdam UMC) according to GLP standard operating procedures [28].

### 2.6. Quantitative Real-Time Polymerase Chain Reaction

Liver and ileum samples stored in RNAlater were homogenized using a MagNA Lyser tissue homogenizer (Roche Applied Science, Penzberg, Germany). Samples were transferred to ceramic bead-containing homogenizer tubes (MagNA Lyser, Green Beads, Roche Applied Science) containing 400 µL of Buffer RLT (RNeasy minikit, Qiagen) and 10 µL/mL of β-mercaptoethanol (Sigma-Aldrich, St. Louis, MO, USA). The samples were disrupted for 60 s (6,500 rpm, RT) to generate tissue homogenates. RNA was isolated from 10 mg liver or ileal tissue using the RNeasy mini kit (Qiagen) according to the manufacturer’s instructions. On-column DNase digestion was performed using an RNase-Free DNase Set (Qiagen) [41]. The RNA concentration was adjusted to 250 ng/µL following spectrophotometric quantification on a NanoDrop (Thermo Fischer Scientific, Waltham, MA, USA). Isolated RNA was stored at −80 °C until further use.

One µg of RNA was reverse transcribed to cDNA with the SensiFAST cDNA Synthesis Kit (Bioline, London, UK) per manufacturer’s instructions. qRT-PCR was performed on a LightCycler 480 (Roche Applied Science) using SensiFAST SYBR No-ROX mix (Bioline). Expression levels of the following liver genes were measured in accordance with de Haan, van der Lely, Warps, Hofsink, Olthof, de Keijzer, Lionarons, Mendes-Dias, Bruinsma, Uygun, Jaeschke, Farrell, Teoh, van Golen, Li and Heger [1]: *Ubc, B2m, Ccnd1* (cyclin D1), *Cdc25b* (M-phase inducer phosphatase 2), *Nr1h4* (FXR)*, Fgfr4* (FGF15 receptor), *Klb* (*β*-Klotho), *Foxm1b, Shp, Stat3, Socs3, Cyp7a1, Cyp8b1, Slc10a1* (NTCP)*, Abcc2* (MRP2), *Abcc3* (MRP3)*, Abcc4* (MRP4), *Abcb11* (BSEP), *Slc51b* (OSTβ), *Tgfb, Slco1a1* (OATP1A1), *Slco1a4* (OATP1A4), *Slco1b2* (OATP1B2), and *Abcb4* (MDR3). For ileum tissue, expression levels of *Hprt, Nr1h4* (FXR), *Fgf15, Abcc3* (MRP3), *Slc10a2* (ASBT), and *Slc51b* (OST*β*) were measured. Primer sequences are provided in Appendix A. Data were processed and analyzed with LinReg software [42] and normalized to the geometric mean of the housekeeping genes *Ubc* and *B2m* (liver samples) or *Hprt* (ileum samples).

### 2.7. Mass Spectrometry

Plasma samples and resected liver tissue samples were analyzed for BAs using UPLC-MS as described [43]. In brief, plasma samples were prepared for analysis using protein precipitation with methanol, and liver samples were lyophilized by 24-h freeze drying. BAs were extracted from dried liver samples using a mixture containing water, acetonitrile, and 2-propanol in a 2:1:1 volume ratio. Samples were pulverized in a Biospec bead beater with 1.0-mm Zirconia beads and centrifuged at 16,000× *g* for 20 min at 4 °C. Quality control samples were prepared from equal parts of plasma supernatant and were used to monitor the performance of the assay. To determine chromatographic retention times of BAs, mixtures of BA reference standards were analyzed following the analysis of study samples. Plasma and liver BA analysis was performed as described [43]. Relative BA intensities were corrected for the liver sample’s dry weight.

### 2.8. Patient Data

Data were gathered from all patients who underwent pre-operative portal vein embolization (PVE) between 2006 and 2017 at the Amsterdam UMC, location AMC. The study was approved by the institutional review board on medical ethics of the Amsterdam UMC, location Academic Medical Center (protocol #W19_252 #19.301, approved on 27 June 2019). The board determined that no patient consent had to be provided for the data curation given the retrospective character of the study. Patients diagnosed with perihilar cholangiocarcinoma who normally undergo biliary drainage before PVE were selected as post-obstructive. Patients with colorectal liver metastasis (CLRM) were selected as “healthy” controls since these patients do not usually suffer from obstructive cholestasis. All patients underwent preoperative assessment of liver function using technetium-99m-labeled mebrofenin hepatobiliary scintigraphy (HBS) in combination with computed tomography (CT) volumetry, as described elsewhere [6,44,45]. These studies were repeated 3 weeks after PVE. All patients underwent embolization of segments 5 to 8, and segments 2 to 4 were designated as the future remnant liver (FRL). The increase in liver function was determined by calculating the percentage increase of the FRL function in %/min/m^2^ compared to the baseline FRL function [46]. The increase in FRL volume was determined by comparing the FRL volume in relation to the total liver volume after PVE to its value before PVE [6].

### 2.9. Statistical Analysis

Statistical analysis was performed in GraphPad Prism 8.0 (GraphPad Software, La Jolla, CA, USA). Continuous variables were expressed as median with inter-quartile-range. A *p*-value of ≤ 0.05 was considered statistically significant. Differences between ordinal variables were analyzed with a Mann–Whitney U test or a Kruskal–Wallis test with Dunn’s multiple comparisons test (Sham-Veh vs. Sham-OCA, BDL-Veh vs. BDL-OCA, and Sham-Veh vs. BDL-OCA). For comparisons concerning the post-obstructive regenerative capacity, a comparison was also made between the Sham-Veh and BDL-Veh groups.

## 3. Results

### 3.1. Obstructive Cholestasis Following Reversible Bile Duct Ligation

The pathological situation of obstructive cholestasis was mimicked by performing rBDL. All animals in the BDL group had high serum and liver tissue BA levels on *t* = −1 day (Figure 2A,B). Moreover, a dramatic increase in serum bilirubin was seen in BDL rats on *t* = −1 day, which normalized after restoration of intestinal bile flow (*t* = 0–5 days, Figure 2C). These results indicate that the rBDL method used here is a reliable method to mimic the pathophysiology of obstructive cholestasis.

### 3.2. Liver Mass Prior to Partial Hepatectomy

Obstructive cholestasis affects gene expression through FXR, among others [1], and mitotic activity of hepatocytes and thus liver growth. Therefore, it is important to investigate the effects of obstructive cholestasis on liver growth in the absence of PHx or OCA administration. Because liver mass is normally maintained at a pre-defined value based on body weight, measured liver weight was corrected for body weight to exclude influences of body weight loss (Appendix A) on liver size. As expected, gross (wet) liver mass was higher in cholestatic rats than in control animals (Sham) 6 days after rBDL and 1 day before PHx (*t* = −1 day) (Figure 3A). In line with previously published results, liver mass in cholestatic (BDL) and control (Sham) groups did not differ significantly after correction for hepatic water content (Figure 3A). Unexpectedly, the number of mitotic cells was significantly higher in the BDL-Veh group compared to the Sham-Veh group (Figure 3B).

### 3.3. Changes in mRNA Expression Prior to Partial Hepatectomy in Cholestatic Rats

Because obstructive cholestasis is characterized by an obstructed biliary tree and impaired bile flow to the ileum, changes in mRNA expression after BDL but before PHx were studied. Decreased ileal *Fgf15* mRNA transcript levels were observed in cholestatic (BDL) rats compared to control animals on *t* = −1 day (Figure 3C, Appendix A). A compensatory change in *Nr1h4*/FXR (Figure 3C) or FXR downstream targets such as *Foxm1* and *Stat3* (Appendix A) mRNA expression was not seen. Proliferation markers *Ccnd1*/cyclinD1 and *Cdc25b*/MPIP2 had unaltered expression levels (Figure 3C). Besides hepatocyte proliferation, FXR governs BA import and export in hepatocytes by regulating the expression of BA transporters. Expression of basolateral importer *Slco1a1*/OATP1A1 was significantly downregulated in BDL rats compared to their control counterparts, whereas the basolateral exporter *Slc51b*/OST*β* was upregulated (Figure 3C, Appendix A).

### 3.4. Liver Regeneration after Partial Hepatectomy

Partial removal of the liver to study liver regeneration is most often performed in healthy rodents. However, liver regeneration in patients with hepatobiliary malignancies is frequently complicated by obstructive cholestasis with accumulation of BAs in the future remnant liver [6,7,8,9]. Excessive BA accumulation and consequent inflammation could impair the regenerative capacity of the liver after PHx dramatically. Therefore, liver regrowth, calculated as the percentage of the original dry liver weight and corrected for body weight, was first compared between post-obstructive (BDL) and non-cholestatic (Sham) rats.

In contrast to previous results, liver regrowth was not impaired in post-cholestatic rats (BDL), in which the bile flow had been restored on *t* = −1 day, compared to control rats (Sham) (Figure 4B). Also, mitosis rates per HPF in the BDL-Veh and Sham-Veh groups were similar at the time of PHx (Figure 4A).

### 3.5. Effects of OCA on Liver Regeneration after PHx

In contrast to the OCA-induced hepatocyte proliferation prior to PHx in cholestatic rats reported previously [24], the liver parenchyma of post-obstructive rats that had been fed OCA (BDL-OCA group) exhibited a similar mitosis rate at the time of PHx (*t* = 0 days) as the BDL-Veh group (Figure 4A). There was no diminished regenerative capacity at later time points (*t* = 1, 3, and 5 days) as reflected by similar liver regrowth in all groups (Figure 4B). The mitosis rate in the BDL-OCA group even exceeded the BDL-Veh group on *t* = 3 days (Figure 4A), but this did not lead to enhanced liver regrowth (Figure 4B).

Besides OCA treatment in post-obstructive rats (BDL-OCA vs. BDL-Veh), the effects of OCA on liver regeneration in non-cholestatic rats were also investigated (Sham-OCA vs. Sham-Veh). Liver regrowth and the number of mitotic cells were higher in Sham-OCA rats compared the Sham-Veh group on *t* = 1 day and *t* = 3 days (Figure 4B). However, these differences were no longer present on day 5.

### 3.6. Changes in Gene Expression after PHx and as Result of OCA Treatment

To elucidate on the effects of OCA on liver growth in post-cholestatic and non-cholestatic rats, PCR was used to demonstrate differences in gene expression as a result of PHx and OCA treatment. *Fgf15* mRNA expression, which is a downstream target of ileal FXR, was higher in OCA-treated groups compared to their Veh-fed counterparts. There was no compensatory downregulation of hepatic *Nr1h4* (encoding FXR) or members of the FGF15 receptor complex *Fgfr4* and *Klb/β*-Klotho (Figure 5A, Appendix A). Enhanced expression of pro-proliferative downstream targets such as *Stat3*, *Foxm1*, and *Cdc25b* (encoding MPIP2) was equally absent except for cyclin D1 in the Sham-OCA group on t = 3 days (Appendix A). Expression of transforming growth factor (TGF)*β* mRNA, which signals termination of liver regeneration, was lower in Sham-OCA rats compared to Sham-Veh rats on *t* = 3 days (Figure 5A).

Besides pro-proliferative signaling, FXR and FGF15 are involved in regulation of BA homeostasis in hepatocytes. The regulation occurs mainly through the downstream target SHP that directly suppresses Na^+^-taurocholate co-transporting polypeptide (NTCP). Regulation is further mediated through other BA receptors and via suppression of BA synthesis enzymes cytochrome P450 (CYP) 7A1 and CYP8B1 [44,47,48], the basolateral exporter organic solute transporter (OST)*β*, and the canalicular exporter bile salt export pump (BSEP) [1,45]. BA-activated FXR also directly induces OST*β* and canalicular exporter multi resistance protein (MRP)2 as well as BSEP complexed with retinoid X receptor [1,49].

Both OCA groups exhibited transcriptional elevation of *Shp* compared to their respective controls, but the follow-up at the level of *Abcb11* (encoding BSEP) did not occur (Figure 5B, Appendix A). Moreover, OCA-treated rats exhibited enhanced expression of *Slc51b/* OST*β* but reduced *Cyp7a1* and *Cyp8b* expression compared to control animals. Expression of other FXR target transporters such as *Abcc2/* MRP2, *Abcb4*/multidrug resistant protein (MDR)3, and *Slc10a1* (NTCP) was similar between OCA-fed groups and controls (Figure 5B and Appendix A), as was intestinal *Slc51b* (OST*β*) expression (Appendix A).

### 3.7. OCA Treatment and Biliary Injury in Post-Cholestatic Rats

Previously, OCA aggravated biliary injury in cholestatic rats as a likely result of forced BA transport into an obstructed biliary tree [27,50]. Therefore, hepatocellular injury related to OCA treatment was also studied in rats after obstructive cholestasis. One day before PHx (*t* = −1 day), ALT and AST levels were higher in (post-)cholestatic (BDL) rats than in sham-operated animals (Figure 6A,B). Similar results were obtained in patients diagnosed with perihilar cholangiocarcinoma, a malignancy that is often complicated by obstructive cholestasis (Table 1). Systemic levels of the biliary injury markers ALP and gGT followed the same dynamics (Figure 6C,D and Table 1). Hepatobiliary injury markers quickly restored and were similar on *t* = 1 day in BDL groups compared to Sham-operated animals. Also, except for gGT on *t* = 3 days, systemic markers were not higher in OCA-treated animals in comparison to their Veh-fed counterparts. These results are contradictory to previous findings where ALP levels remained higher in rBDL-OCA rats compared to rBDL only from *t* = 0 days onward [27].

In addition to hepatocellular injury markers, formalin-fixed liver sections were stained with H & E to assess inflammation, fibrosis, and necrosis (Appendix A) [11]. Livers of rats subjected to BDL showed mild-to-severe purulent cholangitis after PHx (Figure 6E and Appendix A), which was preceded by a minor degree of confluent hepatocellular necrosis that affected <10% of the liver parenchyma. Animals from the BDL groups had higher scores for portal and lobular inflammation (Appendix A). Also, (post-)cholestatic rats lost on average 3.4% of their body weight during 7 days of obstructive cholestasis (*t* = 0 days, Appendix A). At the time of PHx (*t* = 0 days) and after PHx (*t* = 1, 3, and 5 days), livers of post-obstructive (BDL) rats exhibited high portal and lobular inflammation and predominantly showed bridging fibrosis, in contrast to livers of sham-operated animals which only revealed mild periportal fibrosis (Appendix A). Contrary to earlier findings in cholestatic rats [27], OCA treatment did not exacerbate hepatocellular injury when treatment was started after internal biliary drainage.

### 3.8. Bile Acid Pool Composition

Obstructive cholestasis increased serum liver BA levels (Figure 2A,B, Section 3.1). Moreover, a shift from a BA pool predominantly consisting of unconjugated primary BAs, including the potent endogenous FXR agonists chenodeoxycholic acid (CDCA) and the weak agonist cholic acid (CA) [1], towards a pool that mainly consisted of FXR antagonists (tauromuricholic acids) occurred after BDL (Figure 7) [51]. After PHx, the fractional contribution of tauromuricholic acids was reduced again in post-cholestatic rats, whereas the FXR-moot BAs such as most secondary BAs and other conjugated BAs tended to be upregulated in all groups. Upon OCA treatment, the fractional contribution of the tauromuricholic acids was reduced faster than in Veh-treated rats (Figure 7), which is in line with a quicker decrease in serum BAs seen in post-cholestatic rats. As for the liver BA pool (Appendix A), a similar trend in fractional contribution of tauromuricholic acids was observed.

## 4. Discussion

### 4.1. Obstructive Cholestasis Decreases Proliferative Signaling through Intestinal FXR but Does Not Increase Signaling through Hepatic FXR

BAs are synthesized in the liver by, among others, the CYP enzymes CYP7A1, CYP8B1, and CYP27A1. Following production, BAs are secreted into the biliary network and ultimately the duodenum to emulsify dietary lipids and facilitate their digestion by pancreatic enzymes [52,53]. In addition to their role in lipid absorption, BAs act as signaling molecules by for example regulating liver regeneration after PHx [1,23].

BAs activate two pathways that steer hepatocyte proliferation. First, about 95% of the BAs that are secreted into the intestine are reabsorbed by enterocytes [54]. Besides this indirect signaling pathway that is mediated by FGF15/19, BAs can bind hepatic FXR directly. BA-activated hepatic FXR, among other nuclear receptors, also activates FOXM1B [1,55]. In the absence of PHx, physiological bile salt circulation will not lead to liver growth because BAs regulate their own synthesis and transport through a negative feedback loop. BA-activated hepatic FXR inhibits BA synthesis by CYP enzymes through small heterodimer partner (SHP) [1].

Obstructive cholestasis differs from the physiological situation in that it is characterized by an obstructed biliary tree and impaired bile flow to the ileum. Consequently, proliferative signaling through intestinal FXR and FGF15/19 is impeded [56]. Indeed, a strong downregulation of ileal *Fgf15* mRNA transcript levels was observed in cholestatic (BDL) rats compared to control animals on *t* = −1 day. Obstructive cholestasis was also accompanied by increased levels of serum and tissue BA levels, which could theoretically induce proliferative signaling through hepatic FXR. Because BAs differ in toxicity and function, a shift in BA pool composition towards a higher fractional contribution of tauromuricholic acids takes place during cholestasis. Muricholic acids are synthesized from CDCA and conjugated with taurine or glycine residues to enhance serum solubility and reduce membrane permeability [57,58]. Because of this shift, the absolute increase in FXR agonists CDCA and CA compared to FXR antagonists tauro-α/*β*-muricholic acids (T-α/*β*MCA) [51] was lower. This shift in BA pool composition may have been responsible for the absence of hepatic FXR signaling, as evidenced by the unaltered mRNA expression levels of hepatic FXR downstream targets such as *Foxm1* and *Stat3*.

### 4.2. The Increased Liver Weight after BDL Is Not the Result of Proliferation

Previously, an increased wet liver mass in cholestatic (BDL) rats was observed as a result of inflammatory edema and volumetric expansion of the biliary tree one day before PHx. After correction for this hepatic water content, increased liver mass prior to resection was only significant in rats that had been fed OCA during the 7-day BDL period [27]. In the present study, OCA was administered after the 7-day BDL period. For this reason, we investigated the effects of obstructive cholestasis on wet and dry liver mass before PHx in the absence of OCA treatment.

In line with previously published results, liver weight after a 7-day BDL period and 1 day before PHx did not differ significantly after correction for hepatic water content. These results indicate that inflammatory edema and volumetric expansion of the biliary tree are mainly responsible for the increased total liver weight 1 day before PHx. Moreover, the fact that (post-)cholestatic rats lost on average 3.4% of their body weight during 7 days of obstructive cholestasis (*t* = 0 days) suggests a higher energy demand due to inflammation. The noted increase in mitotic cells 1 day before PHx (*t* = −1 day) is difficult to reconcile and cannot be explained by transcriptional analysis of FXR downstream targets *Foxm1* and *Stat3* or markers of proliferation *Ccnd1* (encoding Cyclin D1) and *Cdc25b* (encoding MPIP2). Moreover, the higher number of mitotic cells did not lead to increased liver regrowth at later time points. Possibly, earlier changes in gene expression led to increased mitosis on *t* = −1 day to compensate for the loss of injured hepatocytes, but these changes were minimal and did not lead to more profound liver (re)growth.

### 4.3. Restoration of Intestinal Bile Flow 1 Day before PHx allows for Adequate Regenerative Signaling through FXR after PHx

The unimpaired liver regeneration found in this study is contradictory to published results. Previously, we found that liver regrowth was significantly inhibited 5 days after PHx in post-obstructive rats where bile flow had been restored on *t* = 0 days [27]. Here, we mimicked the human situation by restoring bile flow on *t* = 1 day and found that rats exhibited normal regeneration post-PHx, as was also observed in our patient cohort. There are at least two possible explanations for this finding. First, the restoration of bile flow 1 day before PHx allowed for adequate signaling through ileal FXR and FGF15. Ileal enterocytes express the nuclear BA receptor FXR that, upon BA binding, leads to the ileal expression of FGF15/19 [45,59]. FGF15/19 reaches the liver via the portal circulation and binds to the FGFR4/*β*-Klotho receptor complex. The FGF15/19-activated FGFR4/*β*-Klotho complex induces proliferation through mitogen-activated protein kinase (MAPK) and janus kinase (JAK) following activation of STAT3. The pro-mitogen FOXM1B then leads to proliferation by regulating the timely expression of cyclin-dependent kinases that complex with cyclins such as cyclin D1 [55] and regulate cell-cycle progression by dephosphorylating retinoblastoma protein, a cell cycle inhibitor in phosphorylated form [60,61]. The fact that FGF15, the expression of which was lower on *t* = −1 day in cholestatic rats, was no longer downregulated on *t* = 0–5 days and did not significantly differ between both BDL groups compared to their sham-operated controls suggests that intact signaling through FGF15 and the FGFR4/*β*-Klotho receptor complex is essential for liver regeneration after PHx. This hypothesis is supported by the fact that, in our previous study, the loss of *Fgfr4* after PHx was reported at times of diminished liver regrowth, and the loss of *Fgfr4* was directly correlated with dry liver mass [27]. Here, *Fgfr4* expression was comparable in all groups (Figure 5A) as were the downstream targets *Foxm1* and *Stat3*.

Second, to compare our observation to regeneration in human obstructive livers, clinical data on liver function and growth after PVE were collected from our colorectal liver metastases patient cohort. The model wherein rBDL was relieved 1 day before PHx in rats roughly mimicked the human situation since cholestatic patients normally undergo biliary drainage several weeks before PHx [27]. In line with the regenerative capacity found in rats, no difference was seen in the regenerative capacity of livers after pre-operative PVE as part of the treatment for colorectal liver metastases, a malignancy that is not accompanied by obstructive cholestasis in most cases, and perihilar cholangiocarcinoma that frequently causes cholestasis (Table 1). The fact that both humans and rats where BDL was reversed 1 day before PHx exhibited normal liver regeneration, in contrast to impaired liver regeneration in rats where BDL was only reversed at the time of PHx, indicates that intact signaling through intestinal FXR at the time of PHx is essential for unimpaired liver regeneration.

Finally, the inhibited liver regrowth in our previous study was accompanied by reduced hepatic *Fxr* expression on *t* = 0 days [27]. Hepatic FXR signals proliferation by inducing *Foxm1b* expression directly [55]. In the present study, hepatic *Fxr* expression was not diminished on *t* = 0 days (Figure 5A), attesting to the fact that besides ileal FXR, hepatic FXR is also essential for liver regeneration.

### 4.4. OCA Does Not Affect Liver Regeneration after Obstructive Cholestasis but Accelerates Liver Regeneration in Non-Cholestatic Rats

Besides endogenous FXR agonists such as CDCA, lithocholic acid (LCA), DCA, and CA, exogenous FXR agonist OCA is available for oral administration. Administration of 10 mg/kg OCA is known to accelerate liver regeneration in a rabbit model after PVE without causing elevated aminotransferase levels or histological liver damage [24]. In humans, OCA is indicated as a treatment for primary biliary cholangitis, an auto-immune condition typified by intrahepatic bile duct destruction and cholestasis, although at a lower dosage (5–10 mg daily) [62]. Because of earlier positive results [24,27], a dosage of 10 mg/kg was used in the present study.

Previously, OCA was administered during the period of cholestasis that ranged from 7 day before PHx until 5 days after PHx. Although the 7-day period before PHx led to liver growth in BDL-OCA rats, the livers suffered from a diminished regenerative capacity after PHx. Moreover, OCA administration simultaneously with obstructive cholestasis resulted in aggravated biliary injury and was not fully representative of the clinical situation. In the present study, OCA was therefore administered after cholestasis had been alleviated (i.e., 1 day before PHx until 5 days after PHx).

Although liver regrowth and the actual number of mitotic cells did not increase as a result of OCA treatment in post-obstructive rats (BDL-OCA vs. BDL-Veh), the higher *Fgf15* mRNA expression in OCA-treated groups at all time points indicates proliferative signaling through intestinal FXR. However, the unaltered expression of pro-proliferative downstream hepatocellular targets such as *Stat3*, *Foxm1*, and *Cdc25b* (encoding MPIP2) is consistent with the unaltered regrowth and number of mitotic hepatocytes.

In contrast to OCA treatment in post-obstructive rats, liver regrowth and number of mitotic cells were higher in Sham-OCA rats compared the Sham-Veh group on *t* = 1 day and *t* = 3 days. The results indicate a pro-regenerative effect of OCA in non-cholestatic rats. Corroboratively, *Ccnd1*/CyclinD1 expression was also higher in the Sham-OCA group vs. the Sham-Veh group on *t* = 3 days. However, this difference was no longer observed on day 5 and hence suggests that OCA accelerates liver regrowth but does not increase the ultimate liver volume. In contrast, our previous study in rabbits demonstrated that OCA, dosed at 10 mg/kg from 1 week before to 1 week after PVE, not only augmented liver regrowth but also produced a greater liver volume 7 days after PVE compared to control [24].

The finding that OCA sped up liver regrowth in sham-operated rats but not in cholestatic rats does not seem to be attributable to ileal *Fgf15* and hepatic *Foxm1* (FGFR4 downstream target) mRNA expression profiles on *t* = 0 days and *t* = 1 day, where the differences were not statistically significant between the two groups. Accordingly, factors other than intestinal FXR signaling likely contributed to the accelerated liver regeneration in control rats shortly after PHx. One possible explanation is that lower expression of TGF*β* mRNA in Sham-OCA rats compared to Sham-Veh rats on *t* = 3 days, but not in BDL-OCA rats vs. BDL-Veh rats, could have accounted for the accelerated liver regeneration [63].

### 4.5. OCA Did Not Aggravate Hepatobiliary Injury Because BSEP Was Not Upregulated

Previously it was suggested that OCA aggravated hepatic injury in cholestatic rats because of forced pumping of BAs into an obstructed biliary tree in consequence to induction of the canalicular bile salt export pump (BSEP) by FXR [1,27,64,65,66]. In the present study, obstructive cholestasis was relieved before OCA treatment, allowing for unhindered bile drainage. OCA treatment did not aggravate biliary injury in post-obstructive rats, which may be explained by the similar expression levels of *Abcb11*/BSEP in the OCA-fed and Veh-fed groups. The exact reason behind the absent *Shp*-*Abcb11* response, which is generally expected after oral OCA administration, is currently elusive. One could argue that the changed BA composition in the liver may have abrogated the signaling cascade. However, differential BA composition on *t* = 0 days was only noted in the BDL groups but not the sham-operated animals.

### 4.6. Influence of OCA on BA Homeostasis

The regulation of hepatocellular BA levels is established by three mechanisms. First, the extent of BA synthesis is regulated (Section 4.1). CYP7A1 (cholesterol 7-alpha-hydroxylase) initiates conversion of cholesterol to the primary BAs CA and CDCA, and CYP8B1 regulates the balance between these primary BAs. The expression of CYP7A1 and CYP8B1 is suppressed by SHP, which in turn is induced by BA-activated hepatic FXR and FGF15 [45] to create a negative feedback loop where BAs regulate their own synthesis [44,47,48]. Likewise, the human FGF15 orthologue FGF19 induces SHP through c-Jun N-terminal kinase (JNK) [59], but also increases the stability of SHP by protecting it from proteasomal degradation [67]. OCA treatment led to a lower serum BA concentration on *t* = 0 days after obstructive cholestasis compared to sham-operated rats, possibly resulting from relatively lower *Cyp7a1* and *Cyp8b1* and higher *Shp* mRNA expression in the BDL-OCA group versus BDL-Veh rats at all time points. The same trend was seen in the tissue BA pool, although these differences were not significant.

A second regulatory mechanism of intracellular BA concentration occurs at the level of basolateral and canalicular export. BAs are excreted into the canalicular space by MDR3, MRP2, and BSEP [50,68,69,70,71]. Similarly, BAs are exported into the systemic circulation by MRP3, MRP4, and the OSTα-OST*β* heterodimer [56,68,69,70,71,72,73]. The dynamics of these transporters following OCA treatment were discussed before (Section 3.6). The fact that hepatic *Slc51b* (encoding OST*β*) expression was higher in BDL-OCA rats than in the BDL-Veh group on *t* = 1 day may stem from induced *Slc51b/*OST*β* transcription by OCA-activated hepatic FXR [1,64,65,66]. The similar expression levels of other FXR target genes (*Abcb11* (BSEP), *Abcc2* (MRP2), and *Abcb4* (MDR3) [1,49]) among the OCA-fed animals and controls could possibly be explained by modified expression of other BA receptors. For example, BA-bound FXR induces BSEP expression when complexed with retinoid X receptor (RXR) while RXR, constitutively active androstane receptor, and pregnane X receptor induce MRP2 independently of FXR [1].

A third mechanism that regulates BA homeostasis includes basolateral reabsorption of BAs via NTCP and OATP isoforms (Section 3.7) [69,70,71,74,75,76,77]. The mRNA expression of these transporters was downregulated after PHx. Since this downregulation is similar between OCA-fed groups and Veh-fed counterparts, it is more likely that this downregulation results from PHx-induced release of cytokines, which are known suppressors of hepatic transporters that include NTCP and OATP [78,79] rather than increased exposure to BAs due to redirection of the portal blood flow.

### 4.7. OCA Does Not Affect ICG Clearance and Transporters in Post-Cholestatic and Non-Cholestatic Rats

To qualify liver function in post-obstructive and OCA-treated animals, the ICG-PDR was calculated. ICG is exclusively taken up by the liver and excreted into bile without intrahepatic conjugation, and therefore ICG can be used as quantitative measure for liver function [37,38]. Moreover, the PDR depends on ICG pharmacokinetics and distribution, in particular blood flow to the liver [80]. Taken together, the ICG-PDR can serve as a multifaceted indicator of liver function within its technical limitations [80].

Liver function assessment using ICG-PDR was determined one day after PHx (*t* = 1 day) in all groups and expressed as mean PDR (%/min). In contrast to the histological findings, the mean PDR was similar in all groups (Figure 8), indicating undiminished overall liver function in post-cholestatic rats. Moreover, OCA did not affect liver function in non-cholestatic (Sham-OCA) and post-cholestatic (BDL-OCA) rats as judged by this assessment approach (Figure 8).

ICG clearance is dependent on several parameters, including hepatic blood flow, hepatocellular uptake, and biliary excretion [40,80]. The hepatic transporters NTCP and OATP isoforms are known to be involved in ICG uptake [36]. In contrast, the canalicular exporters responsible for biliary excretion of ICG are still elusive. In line with the mean PDR, mRNA expression of *Slc10a1* (encoding NTCP) and *Slco1a1/1a4/1b2* (encoding OATP isoforms) did not differ significantly between OCA and control groups one day after PHx (*t* = 1 day). Since PDR was similar in all groups, hepatic blood flow was most likely also unperturbed.

## 5. Conclusions

Partial liver resection in patients with perihilar cholangiocarcinoma is associated with high mortality rates [14]. Because of this serious risk, studies have investigated pharmacological modulation of liver regeneration as a means to increase future remnant liver size [15,16,17,18,19,20,21]. OCA is a highly potent exogenous agonist for FXR, and a previous study showed that OCA triggered liver growth prior to hepatectomy in cholestatic rats [27]. Given that OCA was administered during obstructive cholestasis, OCA aggravated biliary injury most likely due to BSEP induction and consequently forced pumping of BAs into an obstructed biliary tree [27]. Moreover, this study did not fully represent the clinical situation. Whereas rats in the initial study underwent biliary drainage simultaneously with PHx, the bile flow in patients with hepatobiliary carcinoma is most often restored by stenting weeks before PHx [29,30].

To emulate the clinical settings, the effects of OCA on liver regeneration and possible side effects in post-obstructive rats were investigated. Despite the fact that OCA administered after a 7-day cholestatic period did not aggravate biliary injury, there was no additional effect on liver (re)growth of daily OCA administration from 1 day prior to PHx onwards in post-cholestatic rats. Previous proliferative effects were associated with strong induction of intestinal *Fgf15* expression after PHx. A persistent induction of *Fgf15* mRNA was found in both OCA-fed, post-obstructive and non-cholestatic rats. However, other proliferation markers remained similar between groups. Moreover, a 6-day period of OCA treatment did not increase the hepatocellular mitosis rate and only accelerated liver regrowth in non-cholestatic rats. Although OCA did not positively affect liver regrowth in post-cholestatic rats, OCA induced a quicker restoration of the serum BA pool after obstructive cholestasis, which could potentially benefit patients. Nonetheless, OCA treatment is also accompanied by significant weight loss in rats, which, if also applicable in humans, could nullify the benefits of slightly higher BA concentrations shortly after PHx. Significant weight loss could delay postoperative recovery due to a catabolic state and therefore lead to more postoperative complications. Reports from the phase III POISE trial dealing with PBC patients, however, did not associate OCA ingestion with weight loss.

Increased liver regrowth in post-cholestatic rats could, in theory, still occur during later time points. A 6-day period (from 1 day prior to PHx to 5 days after PHx) was used because the main purpose of OCA administration was to accelerate liver regrowth so as to minimize postoperative morbidity and mortality, which are highest during the first postoperative days [14]. Previously, proliferative effects of OCA were reported after 7 days and were most pronounced in the rBDL-OCA group, indicating that expansion of liver size only occurs when OCA is administered during obstructive cholestasis and over a longer period. This possibility is supported by the fact that the liver:body weight ratio was not fully restored 5 days after PHx.

In conclusion, the current work challenges the previously reported expected benefits of OCA in post-PHx liver regeneration. These findings highlight the complexity of FXR as a target for intervention in this specific setting and the importance of preclinical studies in clinically representative models. The differences between previous studies and the current work indicate that metabolic conditions such as cholestasis and biliary drainage govern FXR signaling dynamics. Further studies should therefore focus on agents that stimulate liver regeneration but do not exacerbate biliary injury in a setting that mimics the clinical situation as closely as possible.

## Figures and Tables

**Figure 1 biomolecules-11-00260-f001:**
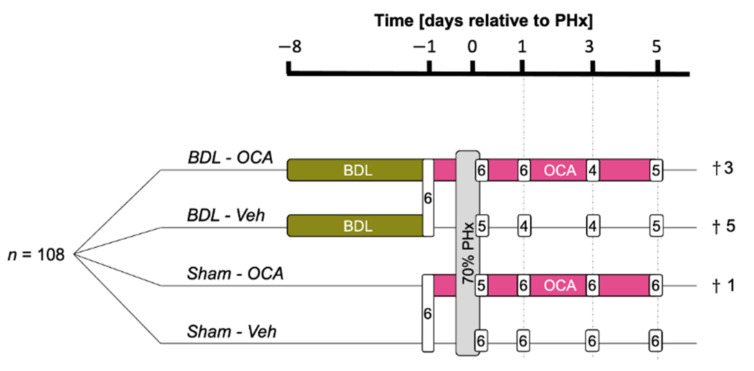
Study design. rBDL, reversible bile duct ligation; Sham, sham surgery; OCA, obeticholic acid; Veh, vehicle; PHx, partial hepatectomy. *n* indicates the number of rats sacrificed at the indicated time point. † refers to animal sacrifice.

**Figure 2 biomolecules-11-00260-f002:**
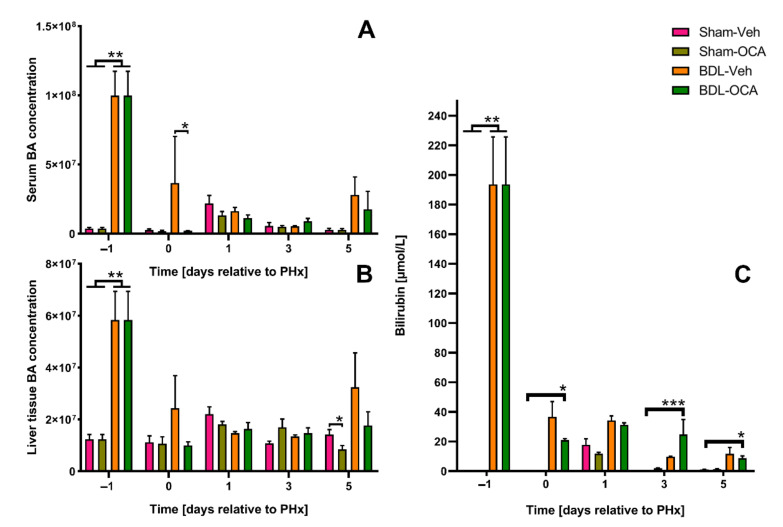
Obstructive cholestasis increases the total amount of serum and liver tissue bile acids (BAs) and bilirubin. Total amount of serum (**A**) and tissue BAs (**B**) and serum bilirubin levels (**C**) 1 day before PHx (*t* = −1 day), on the day of PHx (*t* = 0 days), and on 1, 3, and 5 days after PHx. Data points are given as mean ± SEM. Significance was assessed using a Mann–Whitney U test. * indicates *p* ≤ 0.05; ** indicates *p* ≤ 0.01; *** indicates *p* ≤ 0.001. Veh, vehicle; BDL, bile duct ligation; OCA, obeticholic acid.

**Figure 3 biomolecules-11-00260-f003:**
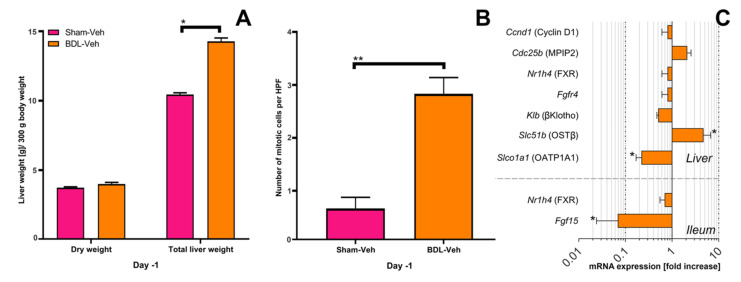
Obstructive cholestasis does not induce proliferation prior to partial resection. (**A**) Dry liver weight and total (wet) liver weight 1 day before PHx (*t* = −1 day) expressed in g per 300 g body weight. (**B**) Number of mitotic cells per high-power field (HPF) in the area with the highest mitotic activity 1 day before PHx (*t* = −1 day). (**C**) mRNA expression levels in liver and ileum in the BDL group 6 days after BDL and 1 day before PHx (*t* = −1 day) expressed as fold-increase relative to the Sham group on *t* = −1 day. Significance was assessed using a Mann–Whitney U test. * indicates *p* < 0.05. Veh, vehicle; BDL, bile duct ligation.

**Figure 4 biomolecules-11-00260-f004:**
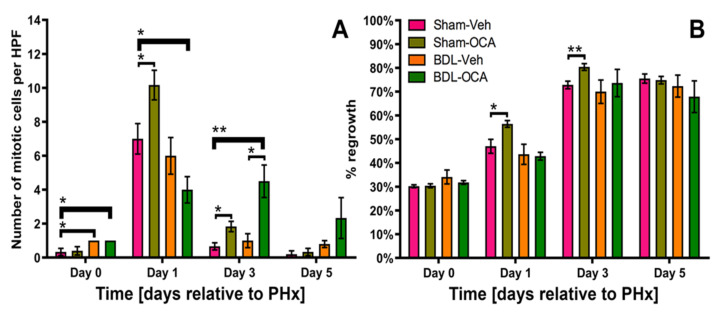
(**A**) Number of mitotic cells per high-power field (HPF) in the area with the highest mitotic activity at the time of PHx (*t* = 0 days) and 1, 3, and 5 days after PHx. (**B**) Liver regrowth, expressed as percentage of dry remnant liver weight of the total dry liver weight, at the time of PHx (*t* = 0 days) and 1, 3, and 5 days after PHx. All data points are provided as mean ± SEM. Significance was assessed using a Mann–Whitney U test. An additional comparison was made between Sham-Veh vs. BDL-Veh. * indicates *p* < 0.05; ** indicates *p* < 0.01. BDL, bile duct ligation; Veh, vehicle; OCA, obeticholic acid.

**Figure 5 biomolecules-11-00260-f005:**
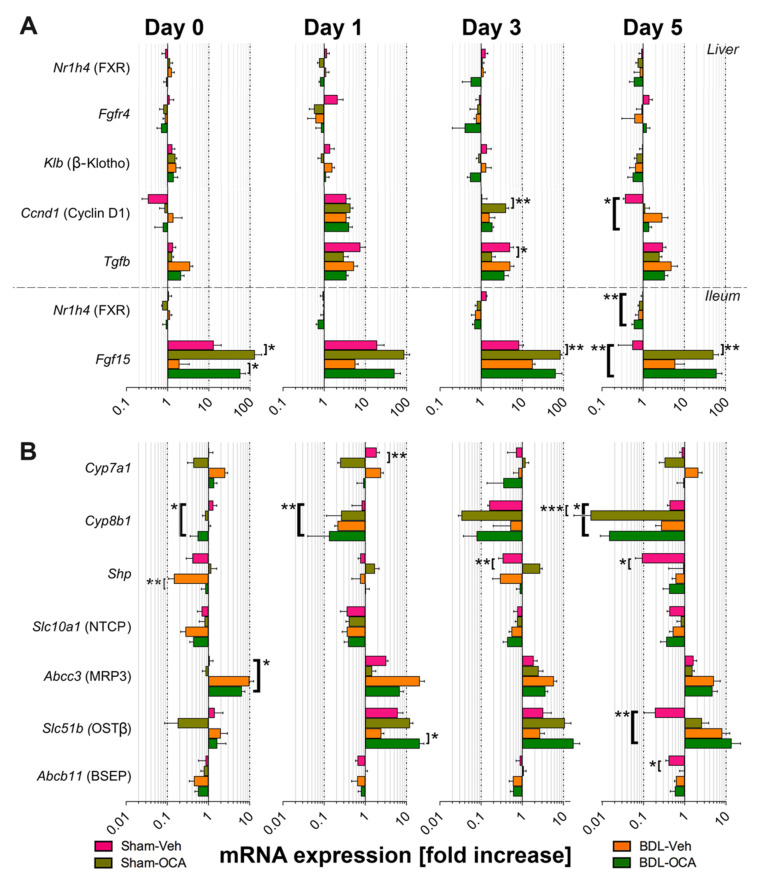
mRNA expression of (**A**) liver and ileum genes involved in liver regeneration and (**B**) liver genes involved in BA homeostasis at the time of PHx (*t* = 0 days) and 1, 3, and 5 days after PHx. Transcript levels are expressed relative to the respective mean transcript levels in control rats on *t* = −1 day. Significance was assessed using a Kruskal–Wallis test with Dunn’s multiple comparisons test. * indicates *p* < 0.05; ** indicates *p* < 0.01; *** indicates *p* < 0.001. All data points represent mean ± SEM. Veh, vehicle; BDL, bile duct ligation; OCA, obeticholic acid.

**Figure 6 biomolecules-11-00260-f006:**
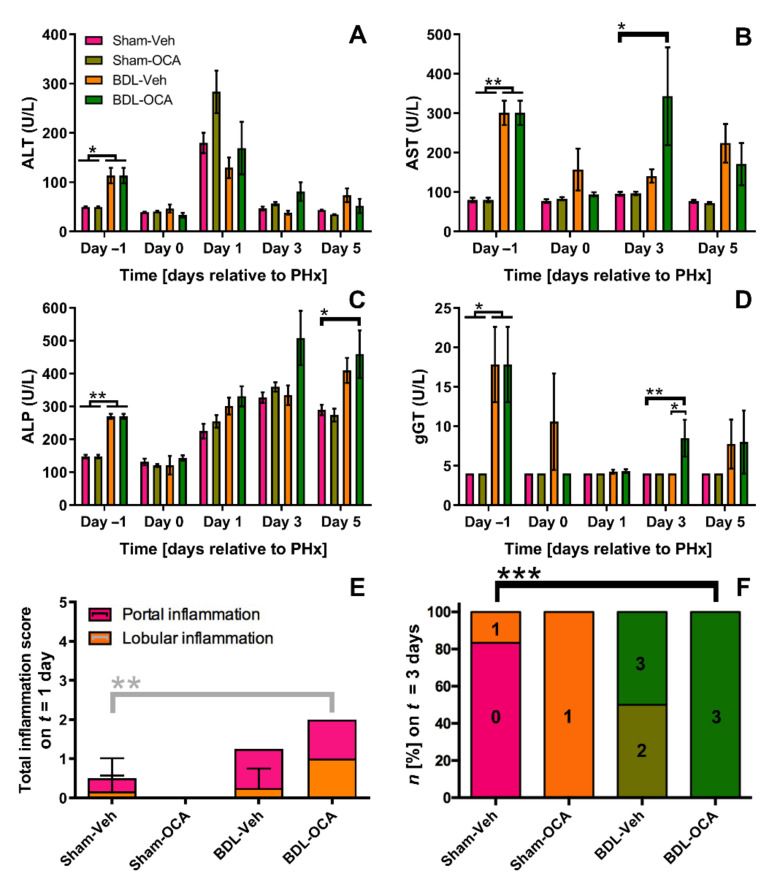
Obeticholic acid does not aggravate hepatobiliary injury caused by BDL. (**A**–**D**) Alanine transaminase (ALT), aspartate transaminase (AST), alkaline phosphatase (ALP), and gamma-glutamyltransferase (gGT) serum levels expressed in units/liter. Data represent mean ± SEM. (**E**) Total inflammation score 1 day after PHx expressed as the sum of the scores for portal inflammation and lobular inflammation (Appendix A). Data represent mean ± SD. (**F**) Types of fibrosis seen in each group 3 days after PHx. 0 = none; 1 = mild periportal fibrosis; 2 = moderate periportal fibrosis; 3 = bridging fibrosis (<50%). Significance was assessed using a Mann–Whitney U test (for *t* = −1 day) or a Kruskal–Wallis test with Dunn’s multiple comparisons test (for *t* = 0, 1, 3, and 5 days). * indicates *p* < 0.05; ** indicates *p* < 0.01; *** indicates *p* < 0.001. BDL, bile duct ligation; Veh, vehicle; OCA, obeticholic acid.

**Figure 7 biomolecules-11-00260-f007:**
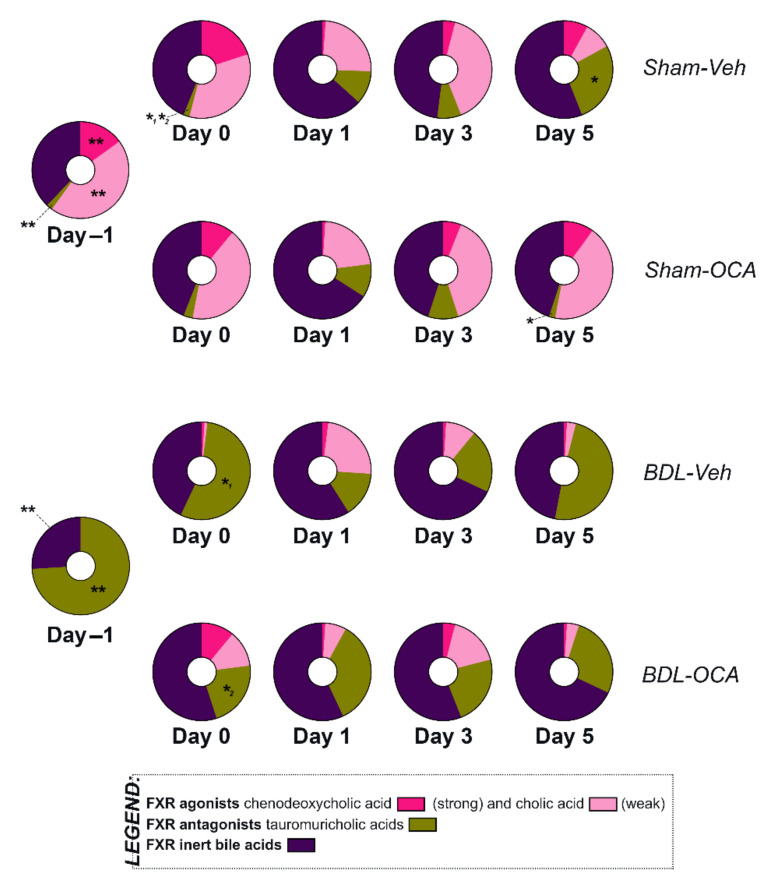
Serum fractional contribution of FXR agonists chenodeoxycholic acid and cholic acid, FXR antagonists tauromuricholic acids, and FXR-inert BAs to the total serum BA pool 1 day before PHx (*t* = −1 day), on the day of PHx (*t* = 0 days), and 1, 3, and 5 days after PHx. Significance in fractional contribution per BA between groups on the same time point was assessed using a Mann–Whitney U test (for *t* = −1 day) or a Kruskal–Wallis test with Dunn’s multiple comparisons test (for *t* = 0, 1, 3, and 5 days). * indicates *p* ≤ 0.05; ** indicates *p* ≤ 0.01. For intergroup analysis: *^1^, Sham-Veh vs BDL-Veh; *^2^, Sham-Veh vs BDL-OCA. BAs, bile acids; Veh, Vehicle; BDL, bile duct ligation; OCA, obeticholic acid.

**Figure 8 biomolecules-11-00260-f008:**
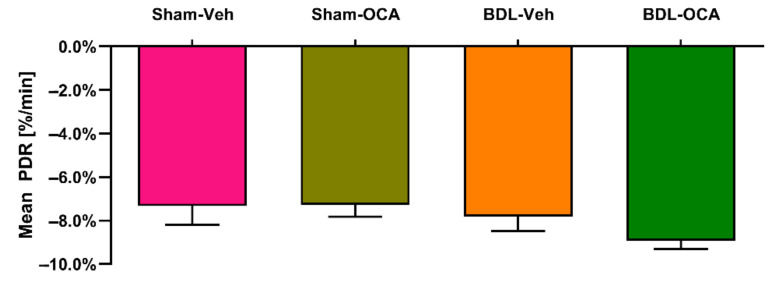
Indocyanine green (ICG) plasma disappearance rate (PDR) expressed as percentage per minute. Significance was assessed using a using a Kruskal–Wallis test with Dunn’s multiple comparisons test. Data points represent mean ± SEM. Veh, vehicle; OCA, obeticholic acid; BDL, bile duct ligation.

**Table 1 biomolecules-11-00260-t001:** Comparison of increases in future remnant liver volume and function after portal vein embolization in patients with perihilar cholangiocarcinoma and colorectal liver metastases.

Parameter	PHC (*n* = 23)	CRLM (*n* = 51)	*p*-Value
**Age**, *median (IQR)*	67 (60–71)	65 (58–69)	0.245
**Male sex**, *n (%)*	16 (70)	34 (67)	0.807
***Days between PVE and CT***, *median (IQR)*	23 (21–26)	22 (21–25)	0.777
**Days between PVE and HBS**, *median (IQR)*	22 (21–24)	22 (21–23)	0.362
**Increase liver volume**, %, *median (IQR)*	35 (15–62)	36 (27–61)	0.392
**Increase liver function**, %, *median (IQR)*	57 (30–89)	54 (35–92)	0.784
**AST**, U/L, *median (IQR)*	69 (36–80)	37 (27–56)	0.133
**ALT**, U/L, *median (IQR)*	73 (52–114)	28 (23–44)	**0.012 ***
**ALP**, U/L, *median (IQR)*	312 (166–620)	121 (77–154)	**0.00 ****
**gGT**, U/L, *median (IQR)*	479 (252–923)	37 (29–108)	**0.00 ****

PHC, perihilar cholangiocarcinoma; CRLM, colorectal liver metastases; PVE, portal vein embolization; CT, computed tomography; HBS, hepatobiliary scintigraphy; AST, aspartate aminotransferase; ALT, alanine aminotransferase; ALP, alkaline phosphatase; gGT, gamma-glutamyltransferase. * indicates *p* ≤ 0.05; ** indicates *p* ≤ 0.01; *p*-values in bold indicate significant differences.

## Data Availability

The data presented in this study are available in the article and Appendix A.

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
