# Peer review of "Unaltered Liver Regeneration in Post-Cholestatic Rats Treated with the FXR Agonist Obeticholic Acid"

_biomolecules, 2021, doi:10.3390/biom11020260_

Round 1
Reviewer 1 Report
This study deals with liver regeneration in cholestatic context, with focus on FXR stimulation. Overall the study builds upon previous work from the same group, providing contradictory results (with their previous paper) and slight increment in knowledge in this very specialized and complex field. The purpose is clinically relevant, stating that major liver surgery has a high mortality rate in cholestatic patients operated for hilar cholangicarcinoma, and that pré-operative FXR stimulation with OCA would induce liver growth, improving post-PH regeneration.
Although the question raised by the authors is highly focalized and the increment in knowledge is very limited, the work is well defined and well performed, and the provided data appear to be solid. However I have some comments and suggestions that may be usefull to improve the manuscript.
- In figure 3, the authors show BA pool composition shifts after BDL and after PH, upon treatment with vehicle or OCA. These data are interesting but not enough commented. The authors should at least explain why these shifts may occur and what is their expected impact on signalling (although this impact is slightly evoked in the text).
- One important point in the manuscript is that intact FXR signalling at the time of PH would be essential for unimpaired regeneration. However the mechanisms involved remain elusive. Authors should give further explanations, going through their gene expression analysis.
- In figure 5C, the authors show "liver regrowth" based on liver mass restoration. Given the very small differences between the groups, it is difficult to be confident only with this readout. Proliferation markers (Ki67 immunohistochemistry...) should be provided in support of the conclusion "Liver regrowth was higher in sham-OCA rats compared the sham-Veh group at t=1 and t=3".
- H and E and Sirius red images should be provided in support of the scores provided in Fig S2, S3, and Figure 6.
- Authors state that Figure 2A shows lower BA in serum upon OCA treatment on t=1. It is not the case. This should be modified.
Reviewer 2 Report
Studies have shown that in rats with obstructive cholestasis due to bile duct ligation, the synthetic bile acid derivative obeticholic acid (OCA) stimulates rat liver regeneration after partial hepatectomy. In contrast to such study designs, handling of patients with obstructive cholestasis, e.g. due to perihilar cholangiocarcinoma, often includes bile drainage before surgical removal of the tumour. In the present study, the authors consequently examined the hypothesis that OCA stimulates liver regeneration in rats after partial hepatectomy performed after opening of a (reversible) 7-days closure of a catheter in the bile duct, i.e. a post-cholestatic model. The results showed that in contrast to findings of OCA induced increased liver regeneration after partial hepatectomy in rats with no bile duct obstruction, OCA did not induce liver regeneration after partial hepatectomy in the post-cholestatic rats. These findings question clinical use of OCA treatment to enhance liver regeneration, liver function, after surgical removal of a tumor causing obstructive cholestasis.
I think it is a clinically relevant interesting study with a fine design. However, the manuscript is pretty difficult to understand. I give two examples and suggest that one or more of the highly experienced international recognized senior scientists revise the manuscript.
- Line 31-33: “In a rat model of liver regeneration, obeticholic acid (OCA) increases liver growth before partial hepatectomy (PHx) through the bile acid receptor farnesoid X-receptor (FXR) when administered during obstructive cholestasis.” This is a very long sentence and difficult to understand: -> “Obeticholic acid (OCA) is used to induce liver growth before partial hepatectomy (PHx) in rats and humans with obstructive cholestasis. This possible takes place through OCA activation of the bile acid receptor farnesoid X-receptor (FXR).”
- Line 81-82: “In a recent study, oral OCA administration for 7 d induced liver growth in cholestatic rats without surgical trigger26. Pre-PHx liver mass was greater in OCA-treated rats than in their control counterparts.” This is a very long sentence and difficult to understand: -> Oral OCA for 7 d induced liver growth in cholestatic rats compared to control rats26..
Specific comments:
- Title should reflect that humans were also studied: Suggestion: Obeticholic acid does not stimulate liver regeneration in rats and humans with obstructive cholestasis. Or, perhaps better, move the presentation of the human studies to the Discussion.
- Line 79: (OCA) is a semi-synthetic: OCA is a synthetic bile acid derivative that is a potent agonist of the nuclear bile acid receptor farnesoid X receptor (FXR).
- Results and discussion: Often not clear what is findings and what is discussion. Should be presented in separate section for clarity. Moreover, for example, the first paragraph does not present or discuss any result of the present study.
- Introduction: Shouldn’t the adverse effects of OCA in patients with PSC be discussed for the present context, and its relevance for the present use of OCA in obstructive cholestasis.
- Line 103: “.. biliary drainage”: How was this performed? Was this controlled for cholestatic effects of the remaining catheter, which is often seen with traditional BDL?
- Introduction last paragraph: This is a mixture of conclusion of the present study and perspectives. I suggest it to be replaced by a clearly formulated background and study hypothesis.
- Line 132: “Rats were randomly assigned to one of 6 groups (Figure 1).” Only 4 groups in the figure?
- Line 150: OCA 10 mg/kg: A huge dose compared to that used in human trials. Does it exceed the transport maximum capacity of the rate-limiting transport BSEP across the canalicular membrane and consequently accumulate in the systemic blood circulation?
- Lines 179-192: Indocyanine green (ICG) is used to measure liver function: The authors use bolus intravenous injection of ICG and calculate the systemic hepatic clearance of ICG from the 10-minute time course of the decay of the plasma concentration of ICG measured in peripheral blood samples. However, this clearance of ICG depends not only on the hepatobiliary excretory capacity, but also on the hepatic blood flow and on the initial distribution of ICG in the systemic blood circulation. Consequently, the present use of ICG is not justified. (See for example Adv Res Gastroenterology and Hepatology 2020;16:555932).
- Could most of the gene data be moved to the Supplementary data - in order to make the key message better understandable?
Round 2
Reviewer 2 Report
Revision fine, thank you.